# Hepatitis E virus exposure and risk factors among ethnic minority populations in Northern Vietnam

Vu Nhi Ha[1], Le Chi Cao[2,3], Tran Hai Dang[4], Dao Thi Huyen[5,6], Nguyen Tien Dung[1], Le Huu Song[2,5,6], Nguyen Linh Toan[5,7], Truong Nhat My[5,6‡]*, Thirumalaisamy P. Velavan[2,5,8]*‡

1 Thai Nguyen University of Medicine and Pharmacy, Vietnam, 2 Institute of Tropical Medicine, University of Tübingen, Tübingen, Germany, 3 Hue University of Medicine and Pharmacy, Hue University, Hue, Vietnam, 4 Thai Nguyen University of Agriculture and Forestry, Vietnam, 5 Vietnamese German Center for Medical Research, VG-CARE, Hanoi, Vietnam, 6 108 Military Central Hospital, Hanoi, Vietnam, 7 Vietnam Military Medical University, Hanoi, Vietnam, 8 Faculty of Medicine, Duy Tan University, Da Nang, Vietnam

‡ These authors are joint senior authors on this work.
* t.velavan@uni-tuebingen.de

## Abstract

### Background

Hepatitis E virus (HEV) causes sporadic outbreaks worldwide, with zoonotic and waterborne genotypes contributing to infections. In Vietnam, HEV genotypes 3 and 4 circulate among humans and swine, but data from remote, ethnic minority populations remain limited.

### Methods

A cross-sectional study was conducted among 272 ethnic minority students at Thai Nguyen University of Medicine and Pharmacy (TUMP) to determine HEV infection markers and associated risk factors. Anti-HEV IgM and IgG were tested in serum samples using Wantai ELISA kits, and HEV RNA was detected by nested PCR targeting the ORF1 region. Demographic and exposure data were collected via structured questionnaires. Statistical analyses were performed using binary logistic regression.

### Results

One participant (0.37%) tested positive for anti-HEV IgM, and 69 (25%) were positive for anti-HEV IgG, while HEV RNA was undetectable. HEV-IgG seroprevalence increased significantly with age (p = 0.004) but showed no sex-related differences. Consumption of tap or mixed water sources (p = 0.043) and raw or undercooked pork liver (p = 0.018) were significantly associated with HEV-IgG positivity. Multivariate analysis confirmed these factors as independent predictors of prior HEV exposure (adjusted OR = 1.6 and 4.8, respectively).

**Data availability statement:** All relevant data are within the paper and its Supporting Information files.

**Funding:** TPV received the funding from PAN-ASEAN Coalition for Epidemic and Outbreak Preparedness (PACE-UP; German Academic Exchange Service (DAAD) Project ID: 57592343). VNH received the funding from Ministry of Education and Training of Vietnam under Project code B2026-TNA-12 (Decision No. 1841/QĐ-BGDĐT).

**Competing interests:** The authors have declared that no competing interests exist.

## Conclusions

A moderate HEV seroprevalence among ethnic minorities indicates substantial prior exposure in northern Vietnam. Strengthening water sanitation, food safety awareness, and routine HEV surveillance is recommended to mitigate infection risk in vulnerable communities.

## Introduction

Hepatitis E virus (HEV) is a major cause of acute viral hepatitis, responsible for millions of infections and significant morbidity in low- and middle-income countries each year [1]. HEV, a member of the family Hepeviridae and belonging to the species *Paslahepevirus balayani*, is a single-stranded RNA virus that can exist in either quasi-enveloped or non-enveloped forms. Transmission primarily occurs via the fecal–oral route through contaminated water in areas with poor sanitation and via foodborne exposure, particularly from pork-derived products. Five HEV genotypes of the virus are known to infect humans. HEV-1 and HEV-2 are primarily transmitted by the faecal–oral route through contaminated water following monsoonal flooding. HEV-1 and HEV-2 genotypes are responsible for large waterborne outbreaks in Asia and Africa and are associated with severe disease in pregnant women, with reported mortality rates ranging from 10% to 30%. In contrast, HEV-3 and HEV-4 genotypes are globally distributed, of zoonotic origin with pigs and wild boar populations as reservoirs [2,3,4]. HEV-7 has been associated with the consumption of camel-derived products in regions such as the Middle East, Pakistan, and parts of Africa [5]. Although these genotypes differ in host range and transmission route, their clinical course and diagnostic profiles follow similar kinetics.

The incubation period of HEV infection ranges from two to nine weeks. HEV RNA can be detected in serum from two to six weeks post-infection and may persist longer in stool. HEV-specific IgM and IgG antibodies typically appear three to four weeks after infection; IgM levels decline within four to six months, whereas IgG antibodies may exists for several years [5]. Diagnosis relies on a combination of serological and molecular assays, with enzyme-linked immunosorbent assays (ELISAs) widely used for the detection of anti-HEV antibodies [5].

In Vietnam, HEV genotypes 3 and 4 circulate among humans and swine, indicating ongoing zoonotic transmission [3,6]. High HEV RNA and antibody prevalence have been reported both in pigs and among individuals with occupational or dietary exposure to animal products [4,7]. However, most studies have focused on urban or peri-urban areas, and data from remote, mountainous regions often inhabited by ethnic minority communities remain limited. A previous survey in Ha Giang Province reported a notably high HEV-IgG seroprevalence of 72.1% [8].

To address this knowledge gap, we investigated HEV infection markers and potential behavioral and environmental risk factors among ethnic minority students at Thai Nguyen University of Medicine and Pharmacy (TUMP). This study provides baseline

evidence of HEV exposure in northern Vietnam and contributes to understanding regional transmission dynamics, supporting future surveillance and prevention strategies in vulnerable populations.

## Materials and methods

### Ethics statement and study population

This study was approved by the Ethics Committee of TUMP (Approval No. P774/ĐHYD-HĐĐĐ). All procedures complied with institutional and national ethical guidelines. Written informed consent was obtained from all participants prior to enrollment. A total of 272 ethnic minority students aged ≥18 years were recruited from TUMP between July–September 2024. Individuals with clinical symptoms of acute hepatitis were excluded. The sex distribution reflects institutional demographics rather than non-random recruitment. All participants were screened and confirmed negative for hepatitis B surface antigen (HBsAg), anti-HCV, anti-HIV, and *Treponema pallidum* antibodies using rapid diagnostic tests. Demographic and exposure-related data, including age, sex, ethnicity, residence, dietary habits, and water source, were collected through structured questionnaires administered by trained study staff.

### Screening of HEV serological markers

Serum samples were tested for anti-HEV IgM and IgG antibodies using commercial enzyme-linked immunosorbent assay (ELISA) kits (Wantai Biopharm, Beijing, China), following the manufacturer's instructions. Absorbance was read at 450 nm with a reference wavelength of 650 nm using an Agilent BioTek Epoch 2 microplate reader (Agilent Technologies, USA). Samples with absorbance values equal to or greater than the assay cut-off were classified as positive. Positive and negative controls supplied with the kit were included in each assay batch to ensure analytical validity.

### Nucleic acid isolation and HEV-RNA detection

Viral RNA was extracted from 140 µL of serum using the QIAamp Viral RNA Mini Kit (Qiagen, Hilden, Germany) according to the manufacturer's protocol and eluted in 60 µL of nuclease-free water. RNA purity and concentration were assessed using a NanoDrop™ spectrophotometer (Thermo Fisher Scientific, Waltham, MA, USA). Extracted RNA was stored at −80 °C until further use. Complementary DNA (cDNA) synthesis was performed using the RevertAid First Strand cDNA Synthesis Kit (Thermo Fisher Scientific, USA) with random hexamer primers. A 307 bp fragment within the ORF1 region of the HEV genome was amplified by nested PCR. The outer PCR used primers HEV-38/HEV-39, and the inner PCR used primers HEV-37/HEV-27, as described previously [7,9]. Each reaction (15 µL total volume) contained 1 × Taq buffer, 200 µM dNTPs, 0.4 µM of each primer, 1 U of Taq DNA Polymerase (Qiagen), and 2 µL of cDNA template. The outer PCR comprised 30 cycles of denaturation at 94 °C for 30 s, annealing at 56 °C for 30 s, and extension at 72 °C for 40 s. The inner PCR included 36 cycles with annealing at 54 °C and extension for 30 s. Positive (previously confirmed HEV RNA sample) and negative (water template) controls were included in every run. PCR amplicons were visualized by 2% agarose gel electrophoresis stained with ethidium bromide and examined under UV illumination.

### Statistical analysis

All the statistical analyses were carried out using the IBM SPSS Statistics version 20.0 (IBM Corp., Armonk, NY, USA). Continuous variables were expressed as medians with interquartile ranges (IQRs), and categorical variables as frequencies and percentages. To compare the groups, chi-square or Fisher's exact tests were used for categorical data and Mann-Whitney U tests for continuous data, depending on data. Binary logistic regression analysis was performed to evaluate the associations between HEV-IgG seropositivity and potential risk factors. The statistical significance was defined as $p < 0.05$.

## Results

### Demographic characteristics of study participants

A total of 272 participants were enrolled, comprising 70% females (n = 191) and 30% males (n = 81). The mean age was 23.6 ± 5.48 years (range: 19–44 years), with most participants (76%) aged between 20 and 29 years (See supplementary data). The cohort represented eight ethnic groups, including Tay (41%), Nung (18%), Thai (11%), Mong (7%), Dao (6%), Muong (6%), San Diu (4%), and others (8%). All participants tested negative for HBsAg, anti-HCV, anti-HIV, and syphilis prior to HEV screening. The detailed baseline characteristics of the study population are summarized in Table 1.

### HEV seroprevalence and RNA positivity

Among the 272 participants, one individual (0.37%) tested positive for anti-HEV IgM, indicating a recent or ongoing infection, whereas 69 individuals (25%) were positive for anti-HEV IgG, suggesting prior exposure. HEV RNA was not detected in any serum samples by nested PCR. The overall HEV-IgG seroprevalence was 25%, consistent with previous findings in urban Vietnamese populations, suggesting considerable exposure to HEV among ethnic minority groups in northern Vietnam.

### Association between HEV seropositivity and demographic factors

HEV-IgG positivity increased significantly with age (p = 0.004). The highest seroprevalence (43%) was observed among participants aged ≥31 years, compared to lower rates in younger groups. No sex-based differences in HEV exposure were observed in adjusted analyses. No statistically significant differences were observed between sexes (male: 35%, female: 21%; p > 0.05). Although the Mong ethnic group exhibited the highest HEV-IgG positivity (50%), followed by Muong (33%) and San Diu (27%), these differences were not statistically significant (p = 0.181). Table 1 illustrates HEV-IgG prevalence by age group, sex, and different ethnicity.

Table 1. Anti-HEV IgG seroprevalence among different ethnicities.

| Characteristics | Variables | anti-HEV IgG seropositivity (n/N), (%) | p-value |
|---|---|---|---|
| Age | ≤ 20 | 6/25 (24) | **0.004** |
| | 21-30 | 46/207 (22) | |
| | ≥ 31 | 17/40 (43) | |
| Sex | Male | 28/81 (35) | p > 0.05 |
| | Female | 41/191 (21) | |
| Ethnicities | Tay | 26/111 (23) | p > 0.05 |
| | Nung | 11/48 (9) | |
| | Thai | 7/31 (23) | |
| | Mong | 9/18 (50) | |
| | Dao | 1/16 (6) | |
| | Muong | 5/15 (33) | |
| | San Diu | 3/11 (27) | |
| | Others# | 7/22 (32) | |

#Cao Lan (5), San Chay (4), San Chi (2) Giay (2), Hoa (2), Ngai (2), Khai (1), La Chi (1), Pa Theu (1), Phu La (1), and Van Kieu (1).

## Behavioral and environmental risk factors

HEV-IgG seropositivity was significantly associated with the type of drinking water source and the consumption of under-cooked pork liver (Table 2). Participants who consumed tap water (30%) or both tap and well water (67%) showed higher seropositivity compared with those who used well water alone (19%) (aOR = 1.6, 95% CI: 1.0–2.4, p = 0.043). Similarly, individuals who reported eating raw or undercooked pork liver had a higher prevalence of HEV-IgG antibodies (64%) than those who did not (24%) (aOR = 4.8, 95% CI: 1.3–16.7, p = 0.018). No significant associations were observed between HEV seropositivity and residence, livestock or wildlife exposure, farming activities, or the consumption of raw or under-cooked wild animal meat, raw seafood, or home-grown vegetables. (p > 0.05) (Table 2).

## Discussion

This study revealed a moderate HEV-IgG seroprevalence of 25% among ethnic minority students in Thai Nguyen Province, northern Vietnam, indicating substantial prior exposure to HEV in this population. The low HEV-IgM positivity (0.4%) and absence of detectable HEV RNA suggest that active infection was rare at the time of sampling. These findings align with previous reports from urban centers such as Hanoi, where comparable IgG seroprevalence rates have been observed among healthy blood donors [10], suggesting that HEV exposure in Vietnam extends beyond high-risk occupational groups to the broader community.

**Table 2. Anti-HEV IgG seropositivity association with selected risk factors.**

| Characteristics | Variables | anti-HEV IgG seropositivity (n/N), (%) | aOR (95% CI) | p-value |
|---|---|---|---|---|
| Drinking water source | Well water | 22/117 (19) | Reference | **0.043** |
| | Tap water | 43/146 (30) | 1.6 (1.0-2.4) | |
| | Both | 2/3 (67) | n/a | |
| Eating raw/undercooked pork liver | Yes | 7/11 (64) | 4.8 (1.3-16.7) | **0.018** |
| | No | 62/261(24) | Reference | |
| Residence | Countryside | 58/212 (27) | 1.5 (0.7-3.2) | p > 0.05 |
| | Urban | 11/55 (20) | Reference | |
| Residence terrain | Hill or mountain | 33/116 (28.4) | 1.4 (0.7-2.6) | p > 0.05 |
| | Delta | 36/161 (22.4) | Reference | |
| Livestock, poultries or wild animals breeding | Yes | 15/58 (26) | 1.3 (0.7-2.3) | p > 0.05 |
| | No | 54/215 (25) | Reference | |
| Exposing to pigs, wild boars or other wild animals | Yes | 21/57 (37) | 1.9 (1.0-3.5) | p > 0.05 |
| | No | 48/215 (22) | Reference | |
| Wild animals hunting | Yes | 4/14 (28.6) | 1.3 (0.4-4.2) | p > 0.05 |
| | No | 65/257 (25.3) | Reference | |
| Farming | Yes | 43/164 (26) | 1.3 (0.7-2.4) | p > 0.05 |
| | No | 26/108 (24) | Reference | |
| Eating raw/undercooked wild animal meat | Yes | 7/11 (64) | 1.6 (0.4-6.2) | p > 0.05 |
| | No | 62/261 (24) | Reference | |
| Eating raw seafood | Yes | 38/147 (25.9) | 1.0 (0.6-1.9) | p > 0.05 |
| | No | 31/125 (24.8) | Reference | |
| Eating home-grown vegetables | Yes | 4/10 (40.0) | 2.9 (0.7-12.5) | p > 0.05 |
| | No | 65/262 (24.8) | Reference | |

aOR: adjusted odds ratio corrected for age and sex.

The observed increase in HEV-IgG seropositivity with age likely reflects cumulative lifetime exposure to contaminated food or water sources. The relatively young age distribution of the cohort may underestimate cumulative lifetime exposure in older community members. Nonetheless, an evident age-dependent escalation in HEV-IgG seropositivity (p = 0.004) was observed, even within this student's cohort. The findings should be interpreted as baseline sero-epidemiological data in young adults of ethnic minority origin, rather than as representative of entire ethnic minority populations. Future studies should incorporate community-based, multi-age sampling frameworks, with a particular focus on remote mountainous provinces.

Despite the study population consisting primarily of university students and reflecting institutional enrolment patterns, with a predominance of female participants, HEV seropositivity did not differ by sex. Analyses were adjusted for age and gender. Nevertheless, it is important to interpret the findings as indicative of young ethnic minority adults, as opposed to the entire community. The necessity of conducting broader population-based studies encompassing various age groups is warranted for future studies. Similar age-dependent patterns have been described in other endemic settings across Asia and Africa [11]. Although seroprevalence was slightly higher among males, the difference was not statistically significant, consistent with other Vietnamese studies reporting no strong sex bias in exposure [10].

Notably, consumption of tap or mixed water sources and raw or undercooked pork liver emerged as independent predictors of HEV exposure. These findings underscore two important transmission pathways: (i) potential contamination of public water systems, possibly by animal or human waste, and (ii) dietary exposure to zoonotic HEV genotypes through undercooked animal products. Dishes such as tiet canh (raw blood pudding) and lightly cooked pork liver are traditional in northern Vietnam and have previously been implicated in HEV transmission [12]. Waterborne outbreaks have been reported in other parts of Asia, particularly involving HEV genotypes 1 and 2 [13,14]. Public-health messaging emphasizing safe water use and thorough cooking of pork products could therefore substantially reduce infection risk.

The HEV-IgG seroprevalence detected in this study was considerably lower than that reported among ethnic minorities in Ha Giang Province (72.1%) [8] and Yunnan, China (66.6%) [15]. Such variation likely reflects differences in environmental sanitation, animal contact frequency, and cultural dietary practices. While pig farming and agricultural work are established risk factors for HEV infection, their association was not significant in this cohort, possibly due to the relatively homogenous student population and limited sample size.

HEV RNA was not detected in any serum samples by nested PCR. HEV RNA detection in this study was performed using a nested PCR assay targeting the ORF1 region, a method previously validated and widely applied. Although more sensitive techniques such as real-time RT-PCR or digital PCR may enhance analytical sensitivity and allow viral load quantification, the absence of detectable HEV RNA in our cohort is biologically plausible. Only one participant (0.37%) demonstrated anti-HEV IgM positivity, indicating very limited recent infection at the time of sampling. Given that HEV viremia is transient and typically detectable for only a short period (2–6 weeks post-infection), RNA negativity is expected in asymptomatic individuals with prior exposure. The relatively high anti-HEV IgG seroprevalence (25%) combined with minimal IgM detection further supports that most infections represented past exposure rather than ongoing viremia. Nevertheless, future studies incorporating stool sampling may provide additional insights on low-level viremia or subclinical infections.

Despite the fact that one participant exhibited a positive result for anti-HEV IgM, HEV RNA was not detected in the serum. This observation does not suggest assay failure or genotype mismatch. The ELISA used in this study has been demonstrated to possess both high sensitivity and specificity, and HEV viremia is only detectable within a limited time window. In view of the established circulation of genotypes 3 and 4 in Vietnam and prior validation of the nested PCR protocol, the absence of RNA most likely reflects sampling outside the viremic phase rather than diagnostic limitations.

The detection of moderate HEV exposure among young adults in a semi-urban academic setting highlights the silent but persistent circulation of HEV in northern Vietnam. The absence of an ethnic majority comparison group limits direct assessment of ethnicity-specific differences in HEV exposure. Future studies incorporating parallel recruitment of majority and minority populations would allow more robust comparative analysis.

Routine diagnostic testing for HEV is also rarely performed in most healthcare facilities, and asymptomatic infections may go unnoticed. Incorporating HEV surveillance into existing hepatitis monitoring programs and improving diagnostic capacity could provide early warning of outbreaks. Water quality monitoring and targeted community education on food safety and hygiene would further support preventive strategies.

In summary, this study provides baseline evidence of HEV exposure among ethnic minority populations in northern Vietnam. The findings point to environmental and dietary routes, particularly unsafe water sources and undercooked pork as key factors influencing HEV transmission. Strengthening water sanitation, promoting food hygiene awareness, and expanding HEV surveillance in both urban and rural communities are critical steps toward reducing the public-health burden of hepatitis E in Vietnam and similar endemic settings.

## Supporting information

**S1 File. Supplementary Data.**
(XLSX)

## Acknowledgments

We acknowledge support from the Open Access Publication Fund of the University of Tübingen.

## Author contributions

**Conceptualization:** Vu Nhi Ha, Thirumalaisamy P. Velavan.

**Formal analysis:** Vu Nhi Ha, Le Chi Cao, Tran Hai Dang, Dao Thi Huyen, Truong Nhat My.

**Funding acquisition:** Thirumalaisamy P. Velavan.

**Investigation:** Vu Nhi Ha, Tran Hai Dang, Dao Thi Huyen, Truong Nhat My.

**Methodology:** Vu Nhi Ha, Tran Hai Dang, Dao Thi Huyen, Truong Nhat My.

**Project administration:** Nguyen Tien Dung, Le Huu Song, Nguyen Linh Toan, Thirumalaisamy P. Velavan.

**Resources:** Nguyen Linh Toan, Thirumalaisamy P. Velavan.

**Supervision:** Nguyen Tien Dung, Le Huu Song, Nguyen Linh Toan, Truong Nhat My, Thirumalaisamy P. Velavan.

**Validation:** Vu Nhi Ha, Tran Hai Dang, Dao Thi Huyen, Truong Nhat My, Thirumalaisamy P. Velavan.

**Writing – original draft:** Vu Nhi Ha, Le Chi Cao.

**Writing – review & editing:** Le Chi Cao, Thirumalaisamy P. Velavan.

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
