## [Decision Letter · Decision Letter 0]

28 Jan 2026

Dear Dr. Velavan,

plosone@plos.org . A letter that responds to each point raised by the academic editor and reviewer(s). You should upload this letter as a separate file labeled 'Response to Reviewers'.A marked-up copy of your manuscript that highlights changes made to the original version. You should upload this as a separate file labeled 'Revised Manuscript with Track Changes'.An unmarked version of your revised paper without tracked changes. You should upload this as a separate file labeled 'Manuscript'.

We look forward to receiving your revised manuscript.

Kind regards,

Massimo Brambilla, Ph.D.

Academic Editor

PLOS One

Journal Requirements:

“The project was funded by the PAN-ASEAN Coalition for Epidemic and Outbreak Preparedness (PACE-UP; German Academic Exchange Service (DAAD) Project ID: 57592343).”

3. We note that your Data Availability Statement is currently as follows: [All relevant data are within the manuscript and its Supporting Information files]

4. Please be informed that funding information should not appear in the Acknowledgments section or other areas of your manuscript. We will only publish funding information present in the Funding Statement section of the online submission form. Please remove any funding-related text from the manuscript.

Reviewers' comments:

Reviewer's Responses to Questions

**Comments to the Author**

1. Is the manuscript technically sound, and do the data support the conclusions?

Reviewer #1: Partly

Reviewer #2: Partly

2. Has the statistical analysis been performed appropriately and rigorously?

Reviewer #1: I Don't Know

Reviewer #2: N/A

3. Have the authors made all data underlying the findings in their manuscript fully available?

Reviewer #1: Yes

Reviewer #2: Yes

4. Is the manuscript presented in an intelligible fashion and written in standard English?

Reviewer #1: Yes

Reviewer #2: No

Reviewer #1: PONE-D-25-59225

Velavan et al. (2025) evaluated HEV exposure and risk factors among ethnic minority populations in northern Vietnam. The issue of Hepatitis E virus (HEV) is highly relevant, particularly in regions with limited access to clean water and sanitation, where zoonotic and waterborne transmission pathways play a significant role in the spread of the virus. Overall, the work is interesting, and it provides valuable insights into the exposure of ethnic minorities in a region where data are limited. However, several aspects need to be clarified.

General comments:

A potential limitation of this study is that the sample consists mainly of university students, who may not be fully representative of the entire ethnic minority population. Students tend to have different socioeconomic and behavioral characteristics, such as better access to healthcare resources and safer food practices, compared to the general population. Additionally, the young age of the sample may not fully reflect long-term exposure to the virus in older age groups. This aspect should be considered when interpreting the results and could be further addressed in future studies with more diverse samples.

A potential limitation of this study is the gender imbalance in the sample, with 70% females (191) and 30% males (81). It is unclear whether this distribution is due to a non-random selection process or reflects a higher proportion of female students at Thai Nguyen University of Medicine and Pharmacy. This gender disparity may affect the generalizability of the findings, as different behaviors or exposures between genders could influence the results.

A further limitation is that only nested PCR was used for detecting HEV RNA. It is possible that using real-time PCR or digital PCR, or targeting a different region of the virus genome, could have identified additional positive cases. The current method may not have been sensitive enough to detect all infections. These limitations, along with others, should be clearly addressed in the discussion section.

Additionally, the content feels somewhat limited in scope. Given the data presented, I would suggest considering whether this could be developed into a short communication instead, as the findings and the dataset may be better suited to this format. With only two tables and a relatively small amount of data, a full-text paper may not be justified. A short communication would allow the authors to present their key findings in a more concise and focused manner.

Reviewer #2: One of the problems of this study is that authors could detect anti-HEV in only one of 272 patients but they could not detect HEV RNA, meaning that 1. This ELISA kit did not work well, or 2. HEV genotype in this patient is different from genotypes 3/4.

1. “Diagnosis relies on a combination of serological and molecular assays, with the WANTAI ELISA widely used for anti-HEV antibody detection.” Need references. Please delete “WANTAI”.

2. Reference [2] seems relatively old. Please refer the following reference: Kanda T, et al. Recent advances in hepatitis E virus research and the Japanese clinical practice guidelines for hepatitis E virus infection.Hepatol Res. 2024 Aug;54(8):1-30. doi: 10.1111/hepr.14062. PMID: 38874115

3. Authors should show the ALT levels of all patients in the present study.

4. Table 2 is incomplete.

5. Authors examined those of ethnic minority students. Authors should include those of ethnic majority students as control.

**Do you want your identity to be public for this peer review?** For information about this choice, including consent withdrawal, please see our Privacy Policy

Reviewer #1: No

Reviewer #2: No

---

## [Author Response · Author response to Decision Letter 1]

24 Feb 2026

Journal Requirements:

“The project was funded by the PAN-ASEAN Coalition for Epidemic and Outbreak Preparedness (PACE-UP; German Academic Exchange Service (DAAD) Project ID: 57592343).” Please state what role the funders took in the study. If the funders had no role, please state: "The funders had no role in study design, data collection and analysis, decision to publish, or preparation of the manuscript." If this statement is not correct you must amend it as needed. Please include this amended Role of Funder statement in your cover letter; we will change the online submission form on your behalf.

Response: The statement is already incorporated in the funding section, as requested.

‘The project was funded by the PAN-ASEAN Coalition for Epidemic and Outbreak Preparedness (PACE-UP; German Academic Exchange Service (DAAD) Project ID: 57592343). The funder has no role in the study design, data collection and analysis, decision to publish, or preparation of the manuscript’.

3. We note that your Data Availability Statement is currently as follows: [All relevant data are within the manuscript and its Supporting Information files]

For example, authors should submit the following data:- The values behind the means, standard deviations and other measures reported;- The values used to build graphs; - The points extracted from images for analysis. Authors do not need to submit their entire data set if only a portion of the data was used in the reported study. If your submission does not contain these data, please either upload them as Supporting Information files or deposit them to a stable, public repository and provide us with the relevant URLs, DOIs, or accession numbers. For a list of recommended repositories, please see https://journals.plos.org/plosone/s/recommended-repositories. If there are ethical or legal restrictions on sharing a de-identified data set, please explain them in detail (e.g., data contain potentially sensitive information, data are owned by a third-party organization, etc.) and who has imposed them (e.g., an ethics committee). Please also provide contact information for a data access committee, ethics committee, or other institutional body to which data requests may be sent. If data are owned by a third party, please indicate how others may request data access.

Response: We confirm that our submission contains all raw data required to replicate the results of the study. The data have been fully de-identified in accordance with institutional ethics approval and applicable data protection regulations. There are no legal or ethical restrictions preventing data sharing.

We respectfully confirm that the Data Availability Statement: “All relevant data are within the manuscript and its Supporting Information files.” is accurate and fully compliant with PLOS ONE data-sharing requirements.

4. Please be informed that funding information should not appear in the Acknowledgments section or other areas of your manuscript. We will only publish funding information present in the Funding Statement section of the online submission form. Please remove any funding-related text from the manuscript.

Response: We confirm that in the submitted version of the manuscript, the Funding Statement and Acknowledgments sections are separate and distinct. All funding-related information has been provided exclusively in the Funding Statement section of the online submission form, in accordance with PLOS ONE guidelines. The Acknowledgments section does not contain funding information.

Response: We have carefully reviewed the publications suggested by the reviewer and evaluated their relevance to our study. Where scientifically appropriate and directly relevant to our work, we have incorporated the suggested citations into the revised manuscript. For references that were not directly relevant to the scope, methodology, or findings of our study, we have not included them, in accordance with the journal’s guidance that citation is not mandatory unless specifically required by the editor.

Reviewers' comments:

Reviewer #1: PONE-D-25-59225

Velavan et al. (2025) evaluated HEV exposure and risk factors among ethnic minority populations in northern Vietnam. The issue of Hepatitis E virus (HEV) is highly relevant, particularly in regions with limited access to clean water and sanitation, where zoonotic and waterborne transmission pathways play a significant role in the spread of the virus. Overall, the work is interesting, and it provides valuable insights into the exposure of ethnic minorities in a region where data are limited. However, several aspects need to be clarified.

General comments:

A potential limitation of this study is that the sample consists mainly of university students, who may not be fully representative of the entire ethnic minority population. Students tend to have different socioeconomic and behavioral characteristics, such as better access to healthcare resources and safer food practices, compared to the general population. Additionally, the young age of the sample may not fully reflect long-term exposure to the virus in older age groups. This aspect should be considered when interpreting the results and could be further addressed in future studies with more diverse samples.

Response: We thank the reviewer for raising this important point. We agree that university students may not fully represent the broader ethnic minority population in northern Vietnam. Our study specifically targeted ethnic minority students enrolled at Thai Nguyen University of Medicine and Pharmacy, as clearly stated in the Methods section . This population was selected for several reasons, the first being the recruitment within a defined academic setting ensured standardized sampling procedures, informed consent, and reliable questionnaire administration. Secondly the ethnic diversity, as the cohort included participants from eight different ethnic minority groups, many originating from remote mountainous areas with historically limited healthcare access. Third, being the public health relevance, Although students may currently reside in a semi-urban academic environment, most were raised in rural settings and retain behavioral and dietary practices (e.g., consumption of traditional pork dishes) reflective of their communities of origin.

This has been discussed now in the revised version as

‘The relatively young age distribution of the cohort may underestimate cumulative lifetime exposure in older community members. Nonetheless, an evident age-dependent escalation in HEV-IgG seropositivity (p = 0.004) was observed, even within this student’s cohort. The findings should be interpreted as baseline sero-epidemiological data in young adults of ethnic minority origin, rather than as representative of entire ethnic minority populations. Future studies should incorporate community-based, multi-age sampling frameworks, with a particular focus on remote mountainous provinces’.

Reviewer comment: A potential limitation of this study is the gender imbalance in the sample, with 70% females (191) and 30% males (81). It is unclear whether this distribution is due to a non-random selection process or reflects a higher proportion of female students at Thai Nguyen University of Medicine and Pharmacy. This gender disparity may affect the generalizability of the findings, as different behaviors or exposures between genders could influence the results.

Response: We appreciate this observation. The gender distribution reflects the actual enrollment structure at Thai Nguyen University of Medicine and Pharmacy during the recruitment period, where female students constitute the majority. There was no intentional gender-based selection.

The study found no statistically significant difference in HEV-IgG seroprevalence between males and females (p > 0.05). Multivariate logistic regression models were adjusted for both age and sex, thus minimising potential confounding effects (see Table 2). The identified independent risk factors (tap/mixed water source and raw/undercooked pork liver consumption) remained significant after adjustment. Consequently, while the gender distribution may compromise strict population representativeness, it did not influence the primary associations observed in this study.

To improve clarity, we will add a statement in the Methods explaining that:

‘The sex distribution reflects institutional demographics rather than non-random recruitment.

in Results section as:

‘No sex-based differences in HEV exposure were observed in adjusted analyses.’

In the discussion as:

‘Despite the study population consisting primarily of university students and reflecting institutional enrolment patterns, with a predominance of female participants, HEV seropositivity did not differ by sex. Analyses were adjusted for age and gender. Nevertheless, it is important to interpret the findings as indicative of young ethnic minority adults, as opposed to the entire community. The necessity of conducting broader population-based studies encompassing various age groups is warranted for future studies.’

Reviewer comment: A further limitation is that only nested PCR was used for detecting HEV RNA. It is possible that using real-time PCR or digital PCR, or targeting a different region of the virus genome, could have identified additional positive cases. The current method may not have been sensitive enough to detect all infections. These limitations, along with others, should be clearly addressed in the discussion section.

Response: We thank the reviewer for this important methodological observation. We agree that real-time RT-PCR or digital PCR platforms may offer enhanced analytical sensitivity and quantitative capability compared with conventional nested PCR approaches.

In the present study, HEV RNA detection was performed using a well-established nested PCR protocol targeting the ORF1 region, incorporating appropriate positive and negative controls. This assay has been previously validated and widely applied in Vietnamese HEV studies, thereby ensuring methodological consistency and comparability with earlier regional data.

Importantly, only one participant (0.37%) tested positive for anti-HEV IgM, indicating very limited recent or active infection at the time of sampling. In this context, the absence of detectable HEV RNA is biologically plausible. HEV viremia is typically transient and detectable only within a relatively short window (approximately 2–6 weeks post-infection). In asymptomatic individuals, RNA negativity is therefore expected even among those with prior exposure. Furthermore, the observed 25% anti-HEV IgG seroprevalence, coupled with minimal IgM detection, supports the interpretation that most infections in this cohort represent past exposure rather than ongoing viremia.

We have discussed in the discussion as below:

‘HEV RNA was not detected in any serum samples by nested PCR. HEV RNA detection in this study was performed using a nested PCR assay targeting the ORF1 region, a method previously validated and widely applied. Although more sensitive techniques such as real-time RT-PCR or digital PCR may enhance analytical sensitivity and allow viral load quantification, the absence of detectable HEV RNA in our cohort is biologically plausible. Only one participant (0.37%) demonstrated anti-HEV IgM positivity, indicating very limited recent infection at the time of sampling. Given that HEV viremia is transient and typically detectable for only a short period (2–6 weeks post-infection), RNA negativity is expected in asymptomatic individuals with prior exposure. The relatively high anti-HEV IgG seroprevalence (25%) combined with minimal IgM detection further supports that most infections represented past exposure rather than ongoing viremia. Nevertheless, future studies incorporating stool sampling may provide additional insights on low-level viremia or subclinical infections.’

Reviewer comment: Additionally, the content feels somewhat limited in scope. Given the data presented, I would suggest considering whether this could be developed into a short communication instead, as the findings and the dataset may be better suited to this format. With only two tables and a relatively small amount of data, a full-text paper may not be justified. A short communication would allow the authors to present their key findings in a more concise and focused manner.

Response: We thank the reviewer for this constructive suggestion. We respectfully believe that the manuscript is appropriately structured as a full original research article rather than a short communication. Although the dataset is moderate in size (n = 272), the study provides comprehensive epidemiological, serological, and molecular data on HEV exposure in a population for which data remain scarce among ethnic minority communities in northern Vietnam. In addition to reporting seroprevalence, the manuscript includes multivariable risk factor analysis, detailed demographic stratification across eight ethnic groups, and contextual interpretation within regional and international HEV epidemiology. These analytical components extend beyond a purely descriptive dataset.

Furthermore, the study integrates laboratory methodology (serology and molecular testing), behavioral risk assessment, and public health implications, offering baseline evidence for HEV surveillance strategies in underserved populations. Given the limited available data from remote mountainous regions, we believe that a full-length format allows for appropriate contextualization, methodological transparency, and discussion of transmission dynamics.

Reviewer #2:

One of the problems of this study is that authors could detect anti-HEV in only one of 272 patients but they could not detect HEV RNA, meaning that 1. This ELISA kit did not work well, or 2. HEV genotype in this patient is different from genotypes 3/4.

Response: We respectfully disagree with the interpretation that the absence of detectable HEV RNA necessarily indicates poor ELISA performance or infection with a different genotype. First, only one participant (0.37%) tested positive for anti-HEV IgM, while 25% were anti-HEV IgG positive. The ELISA kit used (Wantai, Beijing) is internationally validated and widely regarded as one of the most sensitive and specific commercial assays for HEV serology. It has been extensively used in epidemiological studies worldwide and in previous Vietnamese HEV investigations. Internal positive and negative controls were included in each assay batch, ensuring analytical reliability. Therefore, there is no evidence suggesting assay malfunction.

Second, the absence of detectable HEV RNA in the IgM-positive individual is biologically plausible. HEV viremia is transient and typically detectable for only a short period (approximately 2–6 weeks post-infection). In asymptomatic or mildly symptomatic individuals, viral loads may be low and below the detection threshold at the time of sampling. Moreover, th

---

## [Decision Letter · Decision Letter 1]

9 Mar 2026

Hepatitis E virus exposure and risk factors among ethnic minority populations in northern Vietnam

PONE-D-25-59225R1

Dear Dr. Velavan,

We’re pleased to inform you that your manuscript has been judged scientifically suitable for publication and will be formally accepted for publication once it meets all outstanding technical requirements.

Kind regards,

Yury E Khudyakov, PhD

Academic Editor

PLOS One

Additional Editor Comments (optional):

Reviewers' comments:

Reviewer's Responses to Questions

**Comments to the Author**

Reviewer #1: All comments have been addressed

Reviewer #2: (No Response)

2. Is the manuscript technically sound, and do the data support the conclusions?

Reviewer #1: Yes

Reviewer #2: (No Response)

3. Has the statistical analysis been performed appropriately and rigorously?

Reviewer #1: N/A

Reviewer #2: (No Response)

4. Have the authors made all data underlying the findings in their manuscript fully available?

Reviewer #1: Yes

Reviewer #2: (No Response)

5. Is the manuscript presented in an intelligible fashion and written in standard English?

Reviewer #1: Yes

Reviewer #2: (No Response)

Reviewer #1: The authors have satisfactorily addressed all the questions and comments raised during the review process. Their responses were clear and appropriate, and the revisions have improved the overall quality and clarity of the manuscript.

Reviewer #2: All queries have been addressed. This study provides baseline evidence of HEV exposure among ethnic minority

populations in northern Vietnam. The findings point to environmental and dietary routes,

particularly unsafe water sources and undercooked pork as key factors influencing HEV

transmission. Strengthening water sanitation, promoting food hygiene awareness, and expanding HEV surveillance in both urban and rural communities are critical steps toward reducing the public-health burden of hepatitis E in Vietnam and similar endemic settings. Thank you.

**Do you want your identity to be public for this peer review?** For information about this choice, including consent withdrawal, please see our Privacy Policy

Reviewer #1: No

Reviewer #2: No

---

## [Editor Report · Acceptance letter]

PONE-D-25-59225R1

PLOS One

Dear Dr. Velavan,

I'm pleased to inform you that your manuscript has been deemed suitable for publication in PLOS One. Congratulations! Your manuscript is now being handed over to our production team.

Kind regards,

on behalf of

Dr. Yury E Khudyakov

Academic Editor

PLOS One